# Effects of Extracted Pulse Proteins on Lipid Targets for Cardiovascular Risk Reduction: Systematic Review and Meta-Analysis of Randomized Controlled Trials

**DOI:** 10.3390/nu16213765

**Published:** 2024-11-01

**Authors:** Shuting Yang, Songhee Back, Shannan M. Grant, Sabrina Ayoub-Charette, Victoria Chen, Erika J. Lin, Lukas Haintz, Yue-Tong Chen, Elmirah Ahmad, Jacqueline Gahagan, Christopher P. F. Marinangeli, Vanessa Ha, Tauseef Ahmad Khan, Sonia Blanco Mejia, Andreea Zurbau, Russell J. de Souza, Joseph Beyene, Marcia M. English, Vladimir Vuksan, Robert G. Josse, Lawrence A. Leiter, Cyril W. C. Kendall, David J. A. Jenkins, John L. Sievenpiper, Laura Chiavaroli

**Affiliations:** 1Department of Nutritional Sciences, Temerty Faculty of Medicine, University of Toronto, Toronto, ON M5S 1A8, Canada; queenie.yang@mail.utoronto.ca (S.Y.); songhee.back@mail.utoronto.ca (S.B.); sabrina.ayoubcharette@mail.utoronto.ca (S.A.-C.); tori.chen@mail.utoronto.ca (V.C.); erika.lin@mail.utoronto.ca (E.J.L.); yuetong.chen@mail.utoronto.ca (Y.-T.C.); elmirah.ahmad@mail.utoronto.ca (E.A.); tauseef.khan@utoronto.ca (T.A.K.); sonia.blancomejia@mail.utoronto.ca (S.B.M.); andreea.zurbau@mail.utoronto.ca (A.Z.); desouzrj@mcmaster.ca (R.J.d.S.); vuksan@utoronto.ca (V.V.); robert.josse@unityhealth.to (R.G.J.); lawrence.leiter@unityhealth.to (L.A.L.); cyril.kendall@utoronto.ca (C.W.C.K.); david.jenkins@utoronto.ca (D.J.A.J.); john.sievenpiper@utoronto.ca (J.L.S.); 2Toronto 3D Knowledge Synthesis and Clinical Trials Unit, Clinical Nutrition and Risk Factor Modification Centre, St. Michael’s Hospital, Toronto, ON M5B 1W8, Canada; 3Department of Applied Human Nutrition, Faculty of Professional Studies, Mount Saint Vincent University, Halifax, NS B3M 2J6, Canada; shannan.grant2@msvu.ca; 4Department of Obstetrics and Gynaecology, Faculty of Medicine, Dalhousie University, Halifax, NS B3H 4R2, Canada; 5Department of Obstetrics and Gynecology, IWK Health Centre, Halifax, NS B3K 6R8, Canada; 6Department of Medicine, Medical University of Graz, 8010 Graz, Austria; lukas.haintz@stud.medunigraz.at; 7Research Office, Mount Saint Vincent University, Halifax, NS B3M 2J6, Canada; jacqueline.gahagan@msvu.ca; 8Protein Industries Canada, Regina, SK S4P 1Y1, Canada; christopher@proteinsupercluster.ca; 9Department of Clinical Neurosciences, Cumming School of Medicine, University of Calgary, Calgary, AB T2N 1N4, Canada; vha@qmed.ca; 10INQUIS Clinical Research Ltd. (Formerly GI Labs), Toronto, ON M5C 2N8, Canada; 11Department of Health Research Methods, Evidence, and Impact, Faculty of Health Sciences, McMaster University, Hamilton, ON L8S 4L8, Canada; beyene@mcmaster.ca; 12Population Health Research Institute, Hamilton Health Sciences Corporation, Hamilton, ON L8N 1B2, Canada; 13Department of Human Nutrition, Saint Francis Xavier University, Halifax, NS B2G 2W5, Canada; menglish@stfx.ca; 14Division of Endocrinology and Metabolism, Department of Medicine, St. Michael’s Hospital, Toronto, ON M5B 1W8, Canada; 15Department of Medicine, Temerty Faculty of Medicine, University of Toronto, Toronto, ON M5S 1A1, Canada; 16Li Ka Shing Knowledge Institute, St. Michael’s Hospital, Toronto, ON M5B 1W8, Canada; 17College of Pharmacy and Nutrition, University of Saskatchewan, Saskatoon, SK S7N 5E5, Canada

**Keywords:** dietary pulses, extracted proteins, cardiovascular, blood lipids, sex, systematic review, meta-analysis

## Abstract

Background: Many clinical practice guidelines recommend dietary pulses for the prevention and management of cardiovascular disease and diabetes. The impact of extracted pulse proteins remains unclear. We therefore conducted a systematic review and meta-analysis of randomized controlled trials of the effect of extracted pulse proteins on therapeutic lipid targets. Methods and Findings: MEDLINE, Embase, and the Cochrane Library were searched through April 2024 for trials of ≥3-weeks. The primary outcome was low-density lipoprotein-cholesterol (LDL-C). The secondary outcomes were other lipid targets. Independent reviewers extracted data and assessed the risk of bias. Subgroup analyses included by pulse type and the certainty of evidence was assessed using GRADE. Results: Seven included trials (14 trial comparisons, n = 453) with a median of 4-weeks duration and dose of 35 g/day showed that extracted pulse proteins decreased LDL-C by −0.23 mmol/L (95% confidence interval: −0.36 to −0.10 mmol/L, *p* < 0.001). Similar effects were observed for non-high-density lipoprotein-cholesterol and apolipoprotein B. No interactions were found by pulse type. Subgroup analyses revealed effect modification by sex, with greater proportions of females seeing greater reductions. GRADE was generally moderate. Conclusions: Extracted pulse proteins likely result in moderate reductions in LDL-C and other lipid targets. Future studies on various types of extracted pulse proteins including assessments by sex are warranted.

## 1. Introduction

Cardiovascular disease (CVD) is a leading cause of death globally, accounting for 32% of deaths in 2019 [1], costing the healthcare system approximately 30 billion annually in Canada [2,3]. A major risk factor for CVD is low-density lipoprotein-cholesterol (LDL-C) [4,5,6]. Despite advances in drug therapies, those at high CVD risk often have elevated LDL-C due to insufficient lowering with statins, statin-related side effects, poor medication adherence, and treatment inertia [7,8]. National dietary guidelines [9] and cardiovascular, diabetes, and obesity clinical practice guidelines for nutrition therapy [10,11,12,13] recommend dietary patterns with an emphasis on plant-based foods and plant-based protein sources. As a result, the food landscape has rapidly evolved in response to the growing consumer demand for environmentally friendly, plant-based products, stimulating food innovation in the plant-based protein sector. Recent food innovations include an increase in the use of extracted proteins from dietary pulses [14], which are non-oil seeds in the legume family, such as dry peas, chickpeas, lentils, and dry beans. Extracted pulse proteins are often used to manufacture protein supplements or to bolster the nutritional profile of foods. A practical translation of guidelines includes healthcare providers providing actionable dietary advice to their clients about good sources of plant proteins that have evidence of supporting health. However, with the emergence of new products, it is unknown whether there is sufficient evidence for healthcare providers to recommend these products.

Previous studies, which are predominantly based on whole food sources of dietary pulses, have demonstrated beneficial effects on lipid targets of CVD. A previous systematic review and meta-analysis revealed a significant reduction in LDL-C with a median of 130 g, or half a cup, of dietary pulses per day compared to a non-pulse-containing control [15]. With the increased interest in plant-protein products, there has been an emergence of randomized controlled trials investigating the potential cardiovascular benefits of extracted pulse proteins. While the primary component of extracted pulse proteins is protein with modest amounts of starch and fiber, whole pulses also contain higher concentrations of dietary fiber, complex carbohydrates, vitamins, minerals, bioactive compounds, and phytochemicals [16]. Although guidelines recommend consuming dietary pulses, there is a notable gap in understanding whether sources of extracted components of pulses, including protein, can yield similar lipid benefits. To address this knowledge gap and support the development of cardiovascular guidelines, as well as clear and effective recommendations for industry and healthcare professionals and their clients, we conducted a systematic review and meta-analysis of randomized controlled trials to investigate the effect of extracted pulse proteins on established lipid targets for cardiovascular and metabolic syndrome risk reduction [10,17] with an assessment of the certainty of evidence using grading of recommendations, assessment, development, and evaluation (GRADE).

## 2. Materials and Methods

The study followed the *Cochrane Handbook for Systematic Reviews of Interventions* [18] and the Preferred Reporting Items for Systematic Reviews and Meta-Analyses (PRISMA) 2020 (Appendix A) [19]. The study protocol was registered on PROSPERO (CRD42023432826).

### 2.1. Data Sources and Search Strategy

A systemic search was conducted in MEDLINE, Embase, and the Cochrane Central Register of Controlled Trials databases through 30 April 2024. Appendix A present the search strategy. There were no language restrictions. Manual searches of references in the included trials were performed to supplement the database searches.

### 2.2. Study Selection

We included randomized controlled trials in adults of all health backgrounds with intervention periods of at least 3 weeks that investigated the effect of dietary pulse consumption in the form of extracted pulse proteins compared with a non-pulse containing control on LDL-C and other lipids, including non-high-density lipoprotein-cholesterol (non-HDL-C), apolipoprotein B (apoB), high-density lipoprotein-cholesterol (HDL-C), and triglycerides (TGs). We required the follow-up period to be ≥3 weeks, a duration that aligns with the US Food and Drug Administration (FDA) framework for the scientific evaluation of lipid-lowering health claims [20].

Randomized trials in pregnant females or children were excluded. Reports were initially excluded based on a review of their titles and abstracts. The full texts of those reports that remained were reviewed by at least two of the reviewers (SY, SB, VC, SAC, EJL, and YTC) to determine eligibility. In reports containing more than one eligible trial comparison, we included each available trial comparison separately. Reviewer discrepancies were resolved by consensus or arbitration by a senior investigator (LC).

### 2.3. Data Collection and Quality Assessment

Data were extracted using a standardized electronic form by at least two reviewers independently (SY, SB, VC, SAC, EJL, YTC, and SMG). Information included the pulse source (beans, lentils, chickpeas, dried peas, and legume intervention), the number of participants, the participant health status, sex, gender, and mean age, the study design, the energy balance (relative to the background diet), the energy level (intervention relative to the control), setting, comparator, the feeding control, pulse processing, the food form, the macronutrient profile of the diets, the saturated fat and fiber content of the intervention and control groups, the follow-up duration, cholesterol-lowering medication use, the funding source, and the outcome data. All lipid data that were available in publications that indicated lipid outcomes were measured; thus, no authors were contacted for missing outcome data. Graphically presented data were extracted from figures using the Plot Digitizer [21]. Reviewer discrepancies in data extractions were resolved by consensus or arbitration by a senior investigator (LC).

### 2.4. Risk of Bias Assessment

The included studies were assessed for risk of bias (ROB) independently by two investigators using the Cochrane Risk of Bias V.2.0 tool [22]. The assessment was performed across six domains of bias (randomization process, deviations from intended interventions, missing outcome data, measurement of the outcome, selection of the reported result, and overall bias. Crossover studies were assessed for an additional domain of risk of bias arising from period or carryover effects). The ROB for each domain was assessed as “low” (a plausible bias unlikely to seriously alter the results), “high” (a plausible bias that seriously weakens confidence in results), or “some concern” (a plausible bias that raises some doubt about the results). An overall risk of bias was determined based on judgments from each domain. Reviewer discrepancies were resolved by consensus or arbitration by a senior investigator (LC).

### 2.5. Outcomes

The primary outcome was LDL-C determination as the primary lipid target for CVD [10]. Secondary outcomes included the determination of other established lipid targets for CVD (non-HDL-C and apoB) and metabolic syndrome (HDL-C and TG) [17]. Mean differences (MDs) between the intervention and control arm and their standard errors (SEs) were extracted for each eligible trial comparison. If unavailable, they were derived from the available data using published formulas [18]. Mean pairwise differences in change-from-baseline values were preferred over end values, when available. When median data were provided, they were converted to mean data with corresponding variances using methods developed by Luo et al. [23] and Wan et al. [24]. When no variance data were available, the standard deviation was borrowed from a trial similar in size, number of participants, and nature of the intervention, including the food source and dose [25]. When an outcome was not reported, but the variables to calculate that variable were, the outcome was calculated using a standard formula. Non-HDL-C was determined using studies that reported both the total cholesterol and HDL-C levels by calculating the difference between the means. The SDs for non-HDL-C were calculated using the inverse variance law using the SDs of total cholesterol and HDL-C levels [26].

### 2.6. Data Synthesis and Analysis

We used STATA version 17 (StataCorp) for all analyses. Mean pair-wise differences in change from baseline (or end difference) between the extracted pulse proteins group and the non-pulse-containing control group were used as principal effect measurements (significance at *p* < 0.05). The results were expressed as the MDs with 95% confidence intervals (CI). The generic inverse variance method with the DerSimonian and Laird random-effects model was used for data analyses [18,27]. When the number of trials was ≤5, a fixed effects model was used [28]. Paired analyses were applied to all crossover trials with the use of a within-individual correlation coefficient between the treatments of 0.5, as described by Elbourne et al., to calculate the SEs [29,30,31]. To mitigate a unit-of-analysis error, when arms of trials with multiple intervention or control arms were used more than once, the corresponding sample size was divided by the number of times it was used for the calculation of the standard error of the pooled effect [32].

The Cochran Q statistic and the I^2^ statistic were used to assess and quantify heterogeneity, where I^2^ ≥ 50% and P_Q_ < 0.10 represent substantial heterogeneity [18]. Sources of heterogeneity were explored by sensitivity analyses, including individual trial influence, altering the pairwise comparison correlation coefficient, and subgroup analyses. The individual trial influence analysis systematically removed each trial comparison from the meta-analysis with recalculation of the summary effect estimate. A trial whose removal explained the heterogeneity or changed the significance, direction, or magnitude of the effect by more than the minimally important difference (MID) for each outcome (prespecified as 0.1 mmol/L (5%) for LDL-C, non-HDL-C, HDL-C, TG, and 0.04 g/L for apoB) was considered an influential trial [33,34,35,36]. To determine whether the overall results were robust to the use of different correlation coefficients in crossover trials, we also conducted sensitivity analyses using correlation coefficients of 0.25 and 0.75. If ≥10 trials were available, we conducted a priori subgroup analysis to further investigate source heterogeneity using meta-regression (significance at P_Q_ < 0.05) [37,38]. A priori subgroup analyses were conducted by the pulse type, the dose of dietary pulse, the pulse processing method (isolates, concentrates), the food form (food, beverage, mixed, or tablet), the participant health status, sex, gender, and age, cholesterol-lowering medication use, the baseline outcome, the comparator, the duration of follow-up (≤12-weeks, >12-weeks), the study design (crossover, parallel), the energy balance of the intervention relative to the basal diet (positive, neutral, or negative), the feeding control (metabolic, supplemented, or ad libitum), the difference in saturated fat and fiber content between the intervention and control, the type of mean difference (change from baseline, end differences), funding, and the risk of bias domains. Meta-regression analyses were used to assess the significance of each subgroup categorically and, when applicable, continuously. If ≥6 trials are available, generalized least squares trend (GLST) estimation models and spline curve modeling (MKSPLINE procedure) were used to assess linear and nonlinear dose–response relationships [39]. If ≥10 trials are available, then we assessed for the presence of small-study effects (publication bias) by visual inspection of contour-enhanced funnel plots and formal testing with Egger’s and Begg’s tests (significance at *p* < 0.10) [40,41,42]. If there was evidence of small-study effects (publication bias), then we quantified the size of the potential publication bias or other causes of asymmetry by adjusting for the funnel plot asymmetry and assessing the effect of small-study effects using the trim-and-fill method of Duval and Tweedie [43].

### 2.7. Certainty of the Evidence

The certainty of the evidence was assessed using the GRADE approach [44]. The assessments were conducted by 2 independent reviewers (QY, SB), and discrepancies were resolved by consensus or arbitration by the senior author (LC). The evidence was rated as having high, moderate, low, or very low certainty. The included randomized controlled trials were initially rated as high certainty by default and then downgraded or upgraded based on prespecified criteria. The reasons for downgrading the evidence included ROB (assessed by the Cochrane ROB Tool [45]), inconsistency (substantial unexplained interstudy heterogeneity: I^2^ > 50% and P_Q_ < 0.10), indirectness (the presence of factors that limit the generalizability of the results), imprecision (the 95% CI for effect estimates overlap the MID for benefit or harm or lack of robustness from sensitivity analyses), and publication bias (significant evidence of small-study effects). The reason for upgrading the evidence was the presence of a significant dose–response gradient that supports the direction of the pooled effect estimate [46,47,48,49,50,51]. The importance of the magnitude of the pooled estimates was assessed using our prespecified MIDs and the effect size categories according to the GRADE guidance [52,53,54] as follows: a large effect (≥5× MID); moderate effect (≥2× MID); small important effect (≥1× MID); and trivial/unimportant effect (<1 MID).

## 3. Results

### 3.1. Search Results

Figure 1 outlines our systematic search results. We identified 5280 reports from our systematic search, 4881 of which were excluded based on the title or abstract. Of the 399 reports reviewed in full, seven trials met our eligibility criteria. The seven trials provided data on 14 trial comparisons (11 trial comparisons on LDL-C, non-HDL-C, HDL-C; 14 trial comparisons on TG; one trial comparison on apoB) involving 453 participants [55,56,57,58,59,60,61,62].

### 3.2. Trial Characteristics

Table 1 and Appendix A describe the characteristics of the included trials. The trial size had a median of 37 participants (ranging from 24 to 45) for trials of LDL-C, non-HDL-C, and HDL-C; a median of 35 participants (ranging from 14 to 45) for trials of TG; and 38 participants for the trial of apoB. Participants included adults with or without type 2 diabetes or hypercholesterolemia. There were approximately equal ratios of male and female adults. Gender was not reported in any trial. Participants had a median age of 55 y (ranging from 42 to 64 y) for LDL-C, non-HDL-C, and HDL-C, a median age of 54 y (ranging from 42 to 64 y) for TG, and a median age of 57 y (ranging from 54 to 60 y) for apoB. All the trials were conducted in outpatient settings and were performed in Canada (4), the USA (4), Germany (4), Italy (4), Brazil (1), and Australia (1). Most of the trials followed a parallel study design (64% in LDL-C, non-HDL-C, and HDL-C; 71% in TG), except for the apoB outcome, and the feeding control was supplemented. The median dose of extracted pulse proteins was 35 g/day (ranging from 5 to 122 g/day) for LDL-C, non-HDL-C, and HDL-C; 30 g/day (ranging from 1 to 122 g/day) for TG; and 25 g/day for apoB. For most trials, the extracted pulse proteins were produced using a wet extraction method, except for Weiße et al. (2010), Sucher et al. (2017), and Crimarco et al. (2020), where the extraction method was unclear. The extracted pulse proteins were provided in the form of foods (73% in LDL-C, non-HDL-C, and HDL-C; 57% in TG), except for the apoB outcome, where the pulse protein was provided in the form of a beverage. The types of pulses included in trials were beans (55% in LDL-C, non-HDL-C, and HDL-C; 64% in TG, and 100% in apoB), dried peas (36% in LDL-C, non-HDL-C, and HDL-C; 29% in TG; 0% in apoB), and a mix of legumes (9% in LDL-C, non-HDL-C, and HDL-C; 7% in TG; 0% in apoB). The comparators were casein/milk protein (82% in LDL-C, non-HDL-C, and HDL-C; 86% in TG; 100% in apoB) and animal protein (18% in LDL-C, non-HDL-C, and HDL-C; 14% in TG; 0% in apoB), where most indicated that the delivery form and caloric contribution of the control matched that of the intervention. The median follow-up duration was 4 weeks (ranging from 4 to 8 weeks) for LDL-C, non-HDL-C, and HDL-C; and 6 weeks for apoB. Trials were funded by agency sources (64% in LDL-C, non-HDL-C, and HDL-C; 50% in TG; 0% in apoB), followed by industry (27% in LDL-C, non-HDL-C, and HDL-C; 21% in TG; 100% in apoB), and a mix of agency and industry (9% in LDL-C, non-HDL-C, and HDL-C; 29% in TG; 0% in apoB).

### 3.3. Risk of Bias

Appendix A show the risk of bias assessment for individual trials using the Cochrane Risk of Bias Tool 2.0. Across outcomes, most trials were assessed as having a low ROB in outcome measurement (100%), selection domains (100%), and missing outcome domains (91–93%); and some concerns in randomization domains (64–71%) and deviation from the intended intervention (45–57%). Only one trial was assessed as having a high ROB in the missing outcome domain (7–9%) and deviation from the intended intervention (7–9%). Most trials were judged overall as low (50–64%), some trials were assessed as some concerns (27–43%), and one trial was assessed as high (7–9%).

### 3.4. Primary Outcome

Figure 2 and Appendix A show the effect of extracted pulse proteins on LDL-C. Extracted pulse protein consumption resulted in a significant reduction in LDL-C (11 trials; MD −0.23 mmol/L; 95% CI: −0.36 to −0.10; *p* < 0.001) with no substantial heterogeneity (I^2^ = 24.92%; P_Q_ = 0.21)

### 3.5. Secondary Outcomes

Figure 2 and Appendix A present the effect of extracted pulse proteins on non-HDL-C, apoB, HDL-C, and TG. Extracted pulse protein consumption resulted in a significant reduction in non-HDL-C (11 trials; MD = −0.22 mmol/L; 95% CI: −0.36 to −0.08; *p* = 0.002) with substantial heterogeneity (I^2^ = 54.21%; P_Q_ = 0.02) and a significant reduction in apoB (1 trial; MD = −0.16 g/L; 95% CI: −0.19 to −0.13; *p* < 0.001). There was no effect in HDL-C (11 trials; MD = 0.03 mmol/L; 95% CI: −0.00 to 0.07; *p* = 0.076) with no substantial heterogeneity (I^2^ = 0.00%; P_Q_ = 0.91), and no effect in TG (14 trials; MD = −0.03 mmol/L; 95% CI: −0.10, 0.05; *p* = 0.532) with no substantial heterogeneity (I^2^ = 0.00%; P_Q_ = 0.71).

### 3.6. Adverse Events and Acceptability

Appendix A presents the data reported in five trials [55,56,57,58,62] on acceptability and five trials [56,57,58,59,62] on adverse events. Of the five trials reporting on acceptability, participants mainly reported good acceptability of the extracted pulse protein product, except Sirtori et al. (2012) [56] which reported low satisfaction with the consumption of the pulse protein bar. Among the five trials reporting on adverse events, minor gastrointestinal side effects, flatulence, and obstipation were the most reported symptoms, experienced similarly in both the intervention and control groups.

### 3.7. Sensitivity Analyses

Appendix A show the individual trial influence analyses for the effect of extracted pulse proteins. The removal of Bahr et al. 2015 [57] (milk protein) partially explained the substantial heterogeneity (original: I^2^ = 54.21%, P_Q_ < 0.02; after study removed: I^2^ = 49%, P_Q_ = 0.041) and the removal of Frota et al., 2015 [58] fully explained the substantial heterogeneity (original: I^2^ = 54.21%, P_Q_ < 0.02; after study removed: I^2^ = 0%, P_Q_ = 0.503) for non-HDL-C without affecting the magnitude or direction of the effect. The removal of Bahr et al. 2015 [57] (milk protein) resulted in a gain of significance for an increase in HDL-C.

Appendix A shows sensitivity analyses for the different correlation coefficients (0.25 and 0.75) used in paired analyses of crossover trials for each outcome. The use of these different correlation coefficients did not alter the direction, magnitude, or significance of the effect or evidence of substantial heterogeneity.

### 3.8. Subgroup Analyses

Appendix A show categorical and continuous meta-regression analyses for the effect of extracted pulse proteins, where there were at least 10 trial comparisons. There was significant effect modification for the effect of extracted pulse proteins on LDL-C and non-HDL-C by food form, where the one trial that included extracted pulse proteins in beverages showed greater reductions. There was significant effect modification for the effect of extracted pulse proteins on TG by missing outcome reporting, where trials assessed as low ROB showed greater reductions. There was significant effect modification by the proportion of females and males on LDL-C and non-HDL-C, where trials with a greater proportion of females saw a greater reduction. Continuous subgroup analyses across outcomes by proportion of females are summarized in Table 2. In a sensitivity analysis where the one trial which included extracted pulse proteins in beverages, which had the greatest proportion of females (84%) of all included trials (Frota et al., 2015) [58], was removed from the subgroup analyses of sex, there was an attenuation and loss of significant effect modification by sex for LDL-C; however, the effect modification remained significant for non-HDL-C.

### 3.9. Dose–Response Analyses

Appendix A show linear and non-linear dose–response analyses. There was no dose–response for the effect of extracted pulse proteins on LDL-C, non-HDL-C, HDL-C, or TG.

## 4. Small-Study Effects

Appendix A present the contour-enhanced funnel plots and publication bias assessments for all outcomes with ≥10 trials available. There was no evidence of funnel plot asymmetry or publication bias for any outcome.

### GRADE Assessment

Figure 2 and Appendix A show the certainty of evidence assessments by GRADE. The certainty of evidence for the effect of extracted pulse proteins was high for LDL-C (moderate effect); moderate for non-HDL-C (moderate effect), HDL-C (no effect), and TG (no effect), owing to downgrades for imprecision; and low for apoB (trivial effect) owing to a double downgrade for very serious indirectness.

## 5. Discussion

We conducted a systematic review and meta-analysis including 14 trial comparisons providing data on 453 middle-aged adults with or without type 2 diabetes or hypercholesterolemia, assessing the effect of extracted pulse proteins predominantly from beans and dried peas with a median dose of 25–35 g per day over a median follow-up of 4–6 weeks. We showed that extracted pulse proteins resulted in a moderate reduction in LDL-C (−0.23 mmol/L) and non-HDL-C (−0.22 mmol/L), a trivial reduction in apoB (−0.16 mmol/L), and no significant effect on HDL-C and TG. Subgroup analyses of LDL-C and non-HDL-C revealed effect modification by food form, where the one trial using extracted pulse proteins as beverages saw a greater reduction, and by sex, where trials with a greater proportion of females saw a greater reduction. We did not observe a dose–response across outcomes, with most trials providing a narrow dose range between 25 and 35 g/d of extracted pulse proteins, limiting our ability to assess the dose–response gradient.

### 5.1. Findings in Relation to the Literature

Our systematic review and meta-analysis is the first to assess the effect of extracted pulse proteins on established lipid targets. However, a previous systematic review and meta-analysis of whole dietary pulses in 26 trials [15] similarly showed a reduction in LDL-C of 0.17 mmol/L (95% CI: −0.25 to −0.09 mmol/L) with a median dose of 130 g of whole pulses (~12 g protein) per day. However, where we did observe a significant moderate reduction in non-HDL-C of 0.22 mmol/L (95% CI: −0.36 to −0.08); their results were non-significant, yet did tend to show a reduction (−0.09 mmol/L, 95% CI: −0.19 to 0.00 mmol/L). In their subgroup analyses, like our analyses, they did not see significant interaction by pulse types, indicating that all types of dietary pulses behave similarly. Another systematic review and meta-analysis which examined extracted plant proteins which included soy, nuts, and pulses in substitution for animal protein showed a significant reduction in LDL-C of 0.16 mmol/L (95% CI −0.20 to −0.12 mmol/L), non-HDL-C of 0.18 mmol/L (95% CI −0.22 to −0.14 mmol/L), and apoB of 0.05 g/L (95% CI −0.06 to −0.03 g/L), with no subgroup difference by protein type [63]. A further systematic review and meta-analysis on the consumption of extracted soy protein at a median dose of 25 g per day also showed a reduction in LDL-C of 0.12 mmol/L (95% CI −0.17 to −0.072 mmol/L) [64].

Our finding of significant effect modification in the analyses for LDL-C and non-HDL-C by sex is in opposition to that observed in the previous systematic reviews and meta-analyses on whole dietary pulses [15]. Where we saw that trials with a greater proportion of females had greater reductions in LDL-C and non-HDL-C, the previous study found that trials with more males had a greater reduction in LDL-C. In this previous study, the analysis of LDL-C had substantial heterogeneity which was not explained by the subgroup of sex (residual I^2^ = 53%, *p* = 0.01), with no significant effect modification for non-HDL-C where there was substantial heterogeneity (I^2^ = 98%). In contrast, our study had no substantial heterogeneity in the analysis of LDL-C, and for non-HDL-C, the heterogeneity was fully explained by sex (I^2^ = 54% to I^2^ = 0%). The previous study had no other significant subgroups; however, we found that the food form also significantly modified the effect on LDL-C and non-HDL-C. The greater reduction in LDL-C and non-HDL-C observed in the one trial using beverages [58] could partially be explained by the fact that this trial had the greatest proportion of females (84%) among all included trials. As a sensitivity analysis, we removed Frota et al., 2015 [58] from the continuous subgroup analysis by sex and found attenuation and a loss in the significance of effect modification by sex for LDL-C (*p* = 0.101); however, significance was retained for non-HDL-C (*p* = 0.048). The greater reduction in LDL-C and non-HDL-C observed in trials with greater proportions of females could be related to greater adherence to dietary intervention in females and a difference in food preferences between females and males, with females tending to consume more foods including dietary pulses [65]. The sex effect could also be partially explained by females receiving a higher proportion of their protein requirement through the intervention compared to men since the same absolute amount of extracted pulse protein was given to both females and males across all trials. Additionally, the previous study included trials with a median age of 51 y, a baseline LDL-C of 3.5 mmol/L, and a median dose of 130 g whole pulses which provides approximately 12 g protein/day, whereas our trials had a median age of 55 y, baseline LDL-C of 4.1 mmol/L, and a median dose of 35 g/day of extracted pulse protein provided mainly within foods. Future investigations into sex effects in response to pulse proteins should consider factors that may influence sex effects, such as menopausal status for females and baseline LDL-C levels.

There are several mechanisms that may explain the observed effect of extracted pulse proteins on blood lipids. Pulse proteins may alter the gut microbiota composition in hosts, which can affect cholesterol metabolism. Tong et al. showed that mice that were fed pea protein for 30 days had an increased abundance of *Muribaculaceae* and changes in metabolites correlating with reduced LDL-C levels when compared to those that were fed pork protein [66]. Additionally, human studies demonstrate that bioactive peptides from soy proteins, which, like dietary pulses, are considered legumes, may increase hepatic LDL-C receptor expression [67], resulting in an increased clearance of apoB-containing particles from circulation [68]. Furthermore, the mechanism may not be related to changes in body weight since of the nine trials in the present analysis which reported changes in body weight, eight showed reductions in lipid targets independent of body weight.

### 5.2. Strengths and Limitations

Our systematic review and meta-analysis have several strengths. First, we conducted a comprehensive and reproducible search examining the effect of dietary pulses on blood lipids, allowing us to identify effects on extracted pulse proteins. Second, we only included randomized controlled trials which provided evidence that is less susceptible to bias. Third, our meta-analysis had a comprehensive exploration of possible sources of heterogeneity. Fourth, we investigated the shape and strength of dose–response relationships. Fifth, we applied the GRADE approach to assess the certainty of evidence.

Our analysis also has limitations that should be considered when interpreting the results. First, we double downgraded for indirectness for apoB since there was only one trial comparison in predominantly females with hyperlipidemia, which lacks reproducibility and leads to poor applicability of the results to the general adult population. Second, we downgraded for serious imprecision for non-HDL-C, HDL-C, and TG since the 95% CIs of the pooled effect estimate crossed the prespecified MIDs for non-HDL-C and TG, which means that results may not be clinically relevant, and there was a lack of robustness for HDL-C in sensitivity analyses.

Weighing the strengths and limitations, the certainty of evidence was high for LDL-C, moderate for non-HDL-C, HDL-C, and TG, and low for apoB.

### 5.3. Implications

National dietary guidelines [69] and clinical practice guidelines on nutrition therapy for dyslipidemia, CVD, and diabetes [10,70,71] have a focus on plant-based dietary patterns, including plant protein-based foods. Good sources of protein from plants include soy, dietary pulses, nuts, and seeds. Soy and nuts have health claims for cholesterol and coronary heart disease risk reduction [33,72,73], while dietary pulses do not. The observed reduction of 0.23 mmol in LDL-C from fairly high doses (25–35 g/d) of extracted pulse proteins across a variety of food forms and supplements is similar to reductions observed with 25 g/d soy protein [33] and 45 g/d nuts [72,73], as well as with 3 g/day of beta-glucan oat fiber [6,34,74], 7 g/d psyllium [34,74] and 2 g/d plant sterols [5,75], which also carry health claims. The global consumption of dietary pulses is low, at 21 g per capita per day, without change over the past three decades, according to the Food and Agriculture Organization [76]. There is thus an opportunity to increase the population’s intake of dietary pulses for cardiovascular health. The present findings support the use of extracted pulse proteins in nutrient-dense food products in alignment with dietary recommendations. These findings also support healthcare providers in translating guidelines by providing evidence to endorse dietary advice for consuming plant-based protein foods containing extracted pulse proteins for cholesterol reduction, such as plant-based burgers, ground rounds, or plant protein-enriched beverages. However, extracted pulse protein products may not be complete proteins and should be recommended within the context of a dietary pattern that includes a variety of protein foods. This issue may be particularly important in vegetarian and vegan diets for complementarity to ensure the adequate intake of all indispensable amino acids. Traditional sources of pulse proteins, such as whole cooked pulses which are associated with cardiovascular benefit [15,77], not only increase plant protein but also dietary fiber, polyphenols, and other phytonutrients. Thus, ensuring the use of extracted pulse protein products within a dietary pattern high in foods providing other beneficial nutrients, such as dietary fiber, will leverage additive cardiovascular benefits [78].

Additionally, systematic review and meta-analysis assessing the effect of substituting animal for plant protein have demonstrated benefits on blood lipids [63]. Therefore, the use of plant-based protein food sources as substitutes for animal protein foods can support the management of dyslipidemia and reduce the risk of CVD. Moreover, the demonstration of effect modification by sex reinforces the urgent need for the application of sex and gender-based analyses in cardiovascular research to provide evidence to equitably address heart disease, which is driven by underrepresented groups including women [79,80]. Further to our results, emerging evidence underscoring disparities in nutritional behaviors across sexes [65] and the gender spectrum [81,82], including preferences for the intake of dietary pulses [65], drive the call for studies to investigate sex and gender differences in the cardiovascular benefits of pulse intake to inform the tailoring of recommendations in future guideline development.

## 6. Conclusions

In conclusion, our systematic review and meta-analysis identified 14 trial comparisons providing data on 453 middle-aged adults with or without type 2 diabetes or hypercholesterolemia, investigating the effect of extracted pulse proteins on therapeutic lipid targets. The synthesis of evidence from available randomized controlled trials provided a reliable indication that consuming extracted pulse proteins at a mean dose of 35 g per day results in a moderate reduction in LDL-C (−0.23 mmol/L), as well as a good indication for a moderate reduction in non-HDL-C (−0.22 mmol/L) and a trivial reduction in apoB (−0.16 mmol/L), with no effect in HDL-C or TG. The main sources of uncertainty in secondary outcomes were imprecision, as well as indirectness due to only one trial reporting on apoB. To address these uncertainties, there remains a need for larger, high-quality randomized trials assessing a broader variety of pulse types, including chickpeas and lentils, as there were no studies identified on extracted pulse proteins from these pulse types, as well as further exploration of effects by sex and gender. This evidence may direct future policy and guideline updates regarding the use of extracted pulse proteins in food products for the management of cholesterol and the prevention of CVD.

## Figures and Tables

**Figure 1 nutrients-16-03765-f001:**
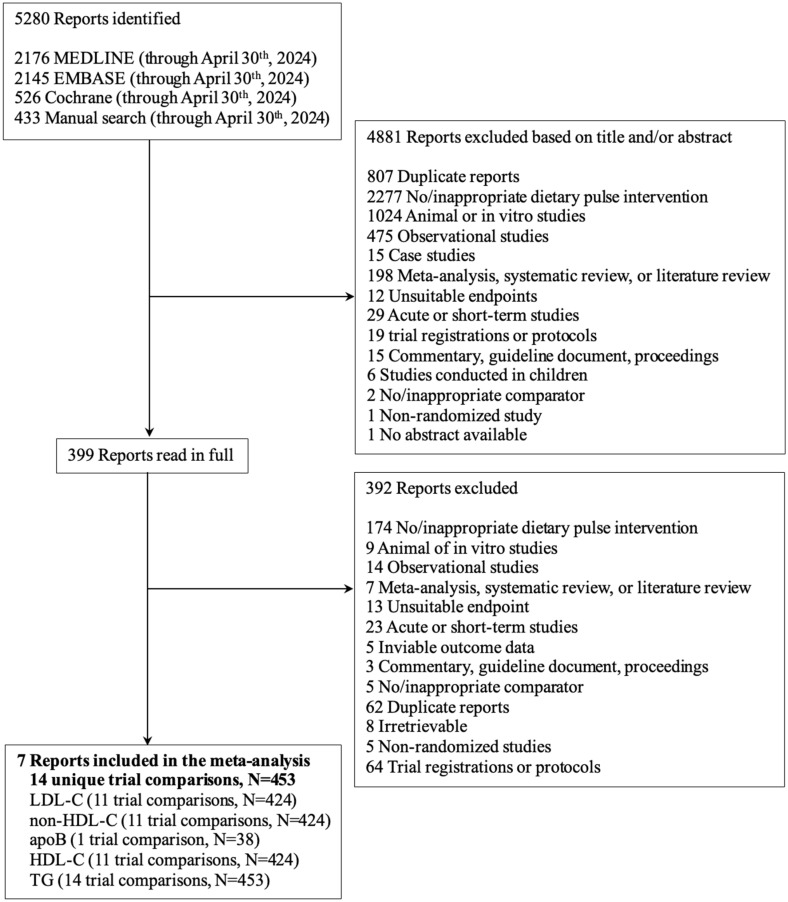
The flow of the literature on the effect of extracted pulse proteins on blood lipids. apoB, apolipoprotein B; HDL-C, high-density lipoprotein cholesterol; LDL-C, low-density lipoprotein cholesterol; non-HDL-C, non-high-density lipoprotein cholesterol; TG, triglyceride.

**Figure 2 nutrients-16-03765-f002:**
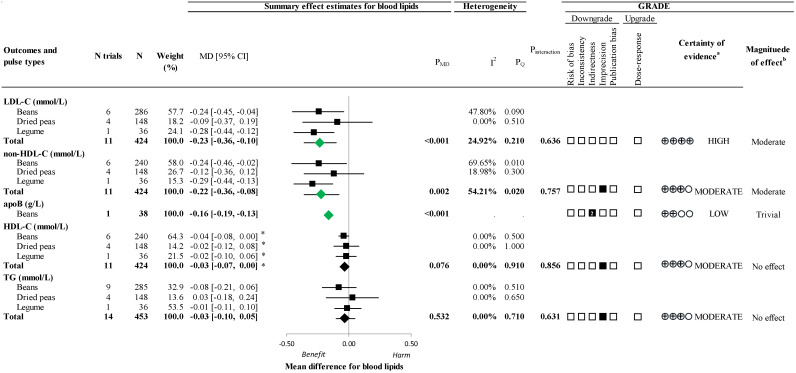
A summary plot of the effect of extracted pulse proteins on blood lipids in randomized controlled trials. Data are expressed as weighted mean differences with 95% confidence intervals of the summary effect estimates using the generic inverse variance method modeled by random effect (≥5 trial comparisons) or fixed effect (<5 trial comparisons) meta-analyses. The between-study heterogeneity was assessed using the Cochran Q statistic, where P_Q_ < 0.100 was considered statistically significant, and quantified by the I^2^ statistic, where I^2^ ≥ 50% was considered evidence of substantial heterogeneity. The effect estimates of total extracted pulse proteins from different sources are denoted as diamonds. The effect estimates of individual extracted pulse protein types are denoted as squares. Any statistically significant reductions are highlighted in green. The grading of recommendations, assessment, development, and evaluation (GRADE) of randomized controlled trials are rated as having a “high” certainty of evidence and can be downgraded by 5 domains and upgraded by 1 domain. The white squares represent no downgrades, filled black squares indicate a single downgrade or upgrade for each outcome, and the black square with a white “2” indicates a double downgrade for each outcome. ^a^ Because all included trials were randomized controlled trials, the certainty of the evidence was graded as high for all outcomes by default and then downgraded or upgraded based on prespecified criteria. Criteria for downgrades included risk of bias (ROB) (downgraded if most trials were considered to be at high ROB); inconsistency (downgraded if there was substantial unexplained heterogeneity: I^2^ ≥ 50%; P_Q_ < 0.10); indirectness (downgraded if there were factors absent or present relating to the participants, interventions, or outcomes that limited the generalizability of the results); imprecision (downgraded if the 95% confidence intervals crossed the minimally important difference (MID) for harm or benefit set at 0.1 mmol/L (5%) for LDL-C, non-HDL-C, HDL-C, and TG and ± 0.04 g/L for apoB [32,33,34,35], or there was a concern with the robustness of the estimate resulting from sensitivity analyses); and publication bias (downgraded if there was evidence of publication bias based on the funnel plot asymmetry and/or significant Egger’s or Begg’s test (*p* < 0.10) with the confirmation of evidence of small study effects by adjustment using the trim-and-fill analysis of Duval and Tweedie [42]). The criteria for upgrades included a significant dose–response gradient that supports the direction of the pooled effect estimate. Please see Appendix A for details on the GRADE assessment. ^b^ For the interpretation of the magnitude, we used the MIDs (see a) to assess the importance of the magnitude of our point estimate using the effect size categories according to the new GRADE guidance [51,52,53] as follows: a large effect (≥5× MID); moderate effect (≥2× MID); small important effect (≥1× MID); and trivial/unimportant effect (<1 MID). Please see Appendix A for details on the GRADE assessment. * Owing to the difference in the directionality of HDL-C compared with the other outcomes with regards to signal for benefit or harm, the sign for the MD was changed. apoB, apolipoprotein B; CI, confidence interval; GRADE, grading of recommendations, assessment, development, and evaluation; HDL-C, high-density lipoprotein cholesterol; LDL-C, low-density lipoprotein cholesterol; MD, mean difference; N, number; non-HDL-C, non-high-density lipoprotein cholesterol; P_MD_, *p*-value of the mean difference; P_Q_, *p*-value of the heterogeneity; ROB, risk of bias; TG, triglycerides.

**Table 1 nutrients-16-03765-t001:** A summary of the characteristics of included trial comparisons assessing the effect of extracted pulse proteins on blood lipids *.

Trial Characteristics	LDL-C	Non-HDL-C	apoB	HDL-C	TG
Trial comparisons (n)	11	11	1	11	14
Study size, median (range) ^a^	37 (24–45)	37 (24–45)	38	37 (24–45)	35 (14–45)
Age (y), median (range)	55 (42–64)	55 (42–64)	57 (54–60)	55 (42–64)	54 (42–64)
Health status (n)	Absence of disease = 2, T2D = 1, Hypercholesterolemia = 8	Absence of disease = 2, T2D = 1, Hypercholesterolemia = 8	Hypercholesterolemia = 1	Absence of disease = 2, T2D = 1, Hypercholesterolemia = 8	Absence of disease = 5, T2D = 1, Hypercholesterolemia = 8
Male:female ratio (%) ^b^	44:59	44:59	16:84	44:59	45:58
Country (No. of comparisons)	Australia = 1, Brazil = 1, Canada = 1, Germany = 4, Italy = 4, USA = 1	Australia = 1, Brazil = 1, Canada = 1, Germany = 4, Italy = 4, USA = 1	Brazil = 1	Australia = 1, Brazil = 1, Canada = 1, Germany = 4, Italy = 4, USA = 1	Australia = 1, Brazil = 1, Canada = 4, Germany = 4, Italy = 4, USA = 4
Study design (%), crossover:parallel	36:64	36:64	100:0	36:64	29:71
Feeding control (%), met:sup:DA:met,sup	0:100:0:0	0:100:0:0	0:100:0:0	0:100:0:0	0:100:0:0
Lipid medication use ratio (%), yes:no:mixed:unclear	0:82:0:8	0:82:0:18	0:100:0:0	0:82:0:18	0:86:0:14
Settings (%), inpatients:outpatients:inpatient,outpatient	0:100:0	0:100:0	0:100:0	0:100:0	0:100:0
Baseline BW (kg), median (range) ^c^	77.4 (66.7–89.5)	77.4 (66.7–89.5)	66.7 (62.2–71.2)	77.4 (66.7–89.5)	81.1 (66.7–89.5)
Baseline BMI (kg/m^2^), median (range)	26.0 (24.7–30.6)	26.0 (24.7–30.6)	27.3 (26.1–28.5)	26.0 (24.7–30.6)	27.3 (24.7–31.5)
Baseline outcome ^d^, median (range)	4.1 (3.1–4.9)	4.9 (3.6–5.6)	1.3 (1.3–1.4)	1.5 (1.1–1.7)	1.5 (1.1–1.8)
Follow-up duration (week), median (range)	4 (4–8)	4 (4–8)	6	4 (4–8)	4 (4–8)
Pulse protein dose (g/day), median (range)	35 (5–122)	35 (5–122)	25	35 (5–122)	30 (1–122)
Intervention and food source (%), extracted and make into a food:beverage:food & beverage:tablet	73:9:9:9	73:9:9:9	0:100:0:0	73:9:9:9	57:7:7:29
Comparator (No. of comparisons)	Animal protein = 2; Casein, milk protein = 9	Animal protein = 2; Casein, milk protein = 9	Casein, milk protein = 1	Animal protein = 2; Casein, milk protein = 9	Animal protein = 2; Casein, milk protein = 12
Energy balance (%), neutral:positive:negative ^e^	91:9:0	91:9:0	100:0:0	91:9:0	71:29:0
Energy control (%), substitution:addition:subtraction ^f^	100:0:0	100:0:0	100:0:0	100:0:0	100:0:0
Funding sources (%), A:I:A,I:NR ^g^	64:27:9	64:27:9	0:100:0:0	64:27:9	50:21:29

A, agency; apoB, apolipoprotein B; BMI, body mass index; BW, body weight; DA, dietary advice; HDL-C, high-density lipoprotein cholesterol; I, industry; LDL-C, low-density lipoprotein cholesterol; met, metabolic; NR, not reported; non-HDL-C, non-high-density lipoprotein cholesterol; sup, supplement; T2D, type 2 diabetes; TG, triglyceride. * All numbers with the exception of baseline values were rounded to the nearest whole number to improve readability. ^a^ All sample sizes reflect participants included in the data analyzed. ^b^ Not all studies reported females and males analyzed. Sitori et al. (2012) [56] reported the number of females and males recruited. ^c^ Not all trials reported baseline values. Baseline values were not reported for baseline BW (n = 5). ^d^ units for LDL-C, non-HDL-C, HDL-C, and TG are in mmol/L and for apoB, g/L. ^e^ Neutral energy balance refers to the maintenance of usual energy intake. A positive energy balance refers to a greater-than-normal energy intake. A negative energy balance refers to a deficit in normal energy intake. ^f^ Energy control refers to the energy intake of the intervention group compared to the control group where substitution refers to energy matched between intervention and comparator, addition refers to excess energy between the intervention and the comparator, and subtraction refers to a deficit in energy between the intervention and the comparator. ^g^ Agency funding is from government, university, or not-for-profit sources. The majority of industry funding is from trade organizations that obtain revenue from the sale of products.

**Table 2 nutrients-16-03765-t002:** Continuous meta-regression analysis for the effect of extracted pulse proteins by the proportion of females *.

Outcome	Female Proportion Range	Trials	Beta [95% CI]	*p*	Residual I^2^ (%)	P_Q_
LDL-C	0.35–0.84	11	−1.53 [−2.47 to −0.59]	0.001	0	0.959
non-HDL-C	0.35–0.84	11	−1.60 [−2.36 to −0.84]	<0.001	0	0.851
HDL-C	0.35–0.84	11	0.15 [−0.12 to 0.43]	0.277	0	0.940
TG	0.35–0.84	14	−0.16 [−0.87 to 0.55]	0.657	0	0.646

Data are presented as the between-group mean difference (95% CI) for a 1-unit change in the predictor variable. β-coefficients were estimated using continuous meta-regression analyses. A positive β-coefficient implies an increase in outcome in the isoflavone intervention as the subgroup variable increases, and a negative β-coefficient implies a decrease in outcome. Residual I^2^ reports inter-study heterogeneity not explained by the subgroup and was estimated using the Cochran Q statistic. CI, confidence interval; HDL-C, high-density lipoprotein cholesterol; LDL-C, low-density lipoprotein cholesterol; non-HDL-C, non-high-density lipoprotein cholesterol; TG, triglyceride. * continuous meta-regression analyses could not be performed for apolipoprotein B (apoB) as there was only one trial comparison.

## Data Availability

The data described in the manuscript, code book, and analytic code will be made available on request.

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
