# Peer review of "Effects of Extracted Pulse Proteins on Lipid Targets for Cardiovascular Risk Reduction: Systematic Review and Meta-Analysis of Randomized Controlled Trials"

_nutrients, 2024, doi:10.3390/nu16213765_

Round 1
Reviewer 1 Report
Comments and Suggestions for Authors
This systematic review and meta-analysis sought to examine the impact of extracted pulse (legume, not soy) proteins on cardiovascular (CVD) risk factors. Included in the meta-analysis were eligible feeding studies that incorporated 35 g/day of extracted pulse proteins into foods and food products such as protein bars, shakes, baked goods and meat alternatives. A previous meta-analysis had examined the effect of 130 g/day of cooked pulses on CVD risk factors, but the increased commercial popularity of food products containing extracted pulse proteins makes this a pertinent topic of inquiry.
The authors carefully and systematically adhered to guidelines for conducting and analyzing data from feeding interventions and the report is accompanied by an extensive list of supplementary materials which provide much valuable additional information. The paper is clear and well-written and the conclusions consistent with the evidence. The major weakness of the review derives from a weakness of the feeding studies themselves and is the fact that most of those studies were made up of the preponderance of female subjects.
Congratulation on your systematically done and good work. One comment relates to Figure 2. The font is too small. Is it possible to display it as landscape?
Author Response
Reviewer 1 - Comment #1. This systematic review and meta-analysis sought to examine the impact of extracted pulse (legume, not soy) proteins on cardiovascular (CVD) risk factors. Included in the meta-analysis were eligible feeding studies that incorporated 35 g/day of extracted pulse proteins into foods and food products such as protein bars, shakes, baked goods and meat alternatives. A previous meta-analysis had examined the effect of 130 g/day of cooked pulses on CVD risk factors, but the increased commercial popularity of food products containing extracted pulse proteins makes this a pertinent topic of inquiry.
The authors carefully and systematically adhered to guidelines for conducting and analyzing data from feeding interventions and the report is accompanied by an extensive list of supplementary materials which provide much valuable additional information. The paper is clear and well-written and the conclusions consistent with the evidence. The major weakness of the review derives from a weakness of the feeding studies themselves and is the fact that most of those studies were made up of the preponderance of female subjects.
Congratulation on your systematically done and good work. One comment relates to Figure 2. The font is too small. Is it possible to display it as landscape?
Response to Reviewer 1 - Comment #1. Thank you for your positive feedback. We have revised Figure 2 to increase the font size and clarity. We have inserted the figure as landscape and will communicate with the Nutrients team to request it be presented as landscape.
Reviewer 2 Report
Comments and Suggestions for Authors
The aim of this meta-analysis is to assess the effects of extracted pulse protein on lipid profiles. The authors included seven published studies, including randomized controlled trials, and concluded that extracted pulse protein may significantly reduce LDL levels.
While this is an interesting meta-analysis, several suggestions for improvement are provided below:
-
It would enhance the clarity of the results to present tables or graphs of the secondary outcomes and subgroup analyses as main figures in the manuscript. This will also benefit the readers.
-
The inclusion of calculated cardiovascular risk score reduction would strengthen the analysis and provide more comprehensive insights into the potential long-term health benefits of extracted pulse protein.
Author Response
Reviewer 2 - Comment #1. The aim of this meta-analysis is to assess the effects of extracted pulse protein on lipid profiles. The authors included seven published studies, including randomized controlled trials, and concluded that extracted pulse protein may significantly reduce LDL levels.
While this is an interesting meta-analysis, several suggestions for improvement are provided below:
It would enhance the clarity of the results to present tables or graphs of the secondary outcomes and subgroup analyses as main figures in the manuscript. This will also benefit the readers.
Response to Reviewer 2 - Comment #1: Thank you for your important feedback. We have both the primary outcome and secondary outcome results presented in Figure 2 and the description of the studies across these outcomes in Table 1. As we conducted multiple subgroup analyses as part of our a priori plan, there are 16 supplemental tables of categorical and 6 tables of continuous subgroup analyses. We therefore have retained the tables of subgroup analyses in the supplemental document. However, we agree that highlighting the main subgroup finding (the continuous meta-regression by proportion of females) in a table within the main manuscript would improve the readability. We have therefore added Table 2.
Reviewer 2 - Comment #2. The inclusion of calculated cardiovascular risk score reduction would strengthen the analysis and provide more comprehensive insights into the potential long-term health benefits of extracted pulse protein.
Response to Reviewer 2 - Comment #2. Thank you for this important comment. The estimated cardiovascular risk score is a valuable calculation to approximate the change in risk based on the change in risk factors in a clinical trial. Unfortunately, none of the trials reported a cardiovascular risk score. The calculation, for example, based on the Framingham Risk Score (https://ccs.ca/app/uploads/2020/12/FRS_eng_2017_fnl1.pdf), requires individual participant data on age, HDL-C, total cholesterol, systolic blood pressure, smoking status and diabetes status. We do not have access to individual data for each trial in order to perform this calculation ourselves in order to estimate the change in cardiovascular risk. Therefore, we are unable to add cardiovascular risk score as an outcome in the present meta-analysis.
Reviewer 3 Report
Comments and Suggestions for Authors
The study investigates the effects of extracted pulse proteins on lipid targets for cardiovascular risk reduction, as dietary pulses are recommended for managing cardiovascular disease and diabetes. A systematic review and meta-analysis of randomized controlled trials were conducted, focusing on trials with a duration of at least 3 weeks. The primary outcome was low-density lipoprotein-cholesterol (LDL-C). The analysis included seven trials with a median duration of 4 weeks and a dose of 35g/day. Extracted pulse proteins significantly decreased LDL-C by -0.23mmol/L and had similar effects on other lipid targets. Extracted pulse proteins likely result in moderate reductions in LDL-C and other lipid targets, with future studies needed to explore different types and effects by sex. Specific comments:
1. The introduction provides a comprehensive background on cardiovascular disease and dietary pulses. However, it could benefit from a more concise summary of the key research question and objectives.
2. The inclusion criteria for the randomized controlled trials are well-defined. Could the authors clarify why trials with intervention periods of less than 3 weeks were excluded?
3. The data extraction process is detailed. Were there any challenges or discrepancies encountered during data extraction, and how were they resolved?
4. The use of the Cochrane Risk of Bias tool is appropriate. Could the authors provide more details on the specific domains where most trials were downgraded?
5. The paper mentions subgroup analyses by pulse type and sex1. Were there any other potential effect modifiers considered that were not included in the final analysis?
6. The results indicate a moderate reduction in LDL-C and other lipid targets. How do these findings compare with previous studies on whole dietary pulses?
7. The GRADE assessment is thorough. Could the authors elaborate on the criteria used for upgrading the evidence based on the presence of a significant dose-response gradient?
8. The discussion highlights the potential for extracted pulse proteins to be recommended in dietary guidelines. What are the practical implications for healthcare providers in terms of dietary advice?
9. The conclusion calls for more high-quality randomized trials. Are there specific types of pulse proteins or populations that should be prioritized in future research?
10. Some of the figures are not labeled properly or are too small to read. I recommend revising these figures to make them more clear and readable.
Author Response
Reviewer 3 - Comment #1. The study investigates the effects of extracted pulse proteins on lipid targets for cardiovascular risk reduction, as dietary pulses are recommended for managing cardiovascular disease and diabetes. A systematic review and meta-analysis of randomized controlled trials were conducted, focusing on trials with a duration of at least 3 weeks. The primary outcome was low-density lipoprotein-cholesterol (LDL-C). The analysis included seven trials with a median duration of 4 weeks and a dose of 35g/day. Extracted pulse proteins significantly decreased LDL-C by -0.23mmol/L and had similar effects on other lipid targets. Extracted pulse proteins likely result in moderate reductions in LDL-C and other lipid targets, with future studies needed to explore different types and effects by sex. Specific comments:
- The introduction provides a comprehensive background on cardiovascular disease and dietary pulses. However, it could benefit from a more concise summary of the key research question and objectives.
Response to Reviewer 3 - Comment #1. Thank you for noting this point to improve clarity of our objectives. We have revised the final paragraph of our introduction to clarify our key research question and objectives. Please see lines 97-101.
- The inclusion criteria for the randomized controlled trials are well-defined. Could the authors clarify why trials with intervention periods of less than 3 weeks were excluded?
Response to Reviewer 3 - Comment #2. Thank you for your comment. We required the follow-up period to be ≥3 weeks, a duration that aligns with the US Food and Drug Administration (FDA) framework for the scientific evaluation of lipid-lowering health claims. We have added a sentence in the methods to include this justification. Please see lines 123-124.
- The data extraction process is detailed. Were there any challenges or discrepancies encountered during data extraction, and how were they resolved?
Response to Reviewer 3 - Comment #3. Thank you for highlighting this missing information. We have added lines 144-145 under Data Collection to clarify:
“Reviewer discrepancies in data extractions were resolved by consensus or arbitration by a senior investigator (LC).”
- The use of the Cochrane Risk of Bias tool is appropriate. Could the authors provide more details on the specific domains where most trials were downgraded?
Response to Reviewer 3 - Comment #4. Thank you for your comment. There is only 1 trial which was rated as having high risk of bias in the missing outcome domain and deviation from intended intervention. This was because there was over 20% missing data (23%) due to non-compliance with the protocol (albeit equally across the intervention and control groups). We have revised the manuscript to clarify there was only one trial with high risk of bias (please see lines 295-298). Since most trial comparisons were rated as low risk of bias or some concerns, with only the one trial rated as high risk of bias, we did not downgrade our GRADE assessment for risk of bias across outcomes.
- The paper mentions subgroup analyses by pulse type and sex1. Were there any other potential effect modifiers considered that were not included in the final analysis?
Response to Reviewer 3 - Comment #5. Thank you. We included multiple subgroup analyses in order to assess for potential effect modification. These are described in our methods, lines 204-214:
“A priori subgroup analyses were conducted by pulse type, dose of dietary pulse, pulse processing (isolates, concentrates), food form (food, beverage, mixed, tablet), participant health status, sex, gender, age, cholesterol-lowering medication use, baseline outcome, comparator, duration of follow-up (≤12-weeks, >12-weeks), study design (crossover, parallel), energy balance of the intervention relative to the basal diet (positive, neutral, negative), feeding control (metabolic, supplemented, ad libitum), difference in saturated fat and fiber content between the intervention and control, type of mean difference (change from baseline, end differences), funding, and risk of bias domains. Meta-regression analyses were used to assess the significance of each subgroup categorically and, when applicable, continuously.”
With regards to results, only sex was found to be significant with a greater proportion of females showing greater reductions in LDL-C and non-HDL-C.
- The results indicate a moderate reduction in LDL-C and other lipid targets. How do these findings compare with previous studies on whole dietary pulses?
Response to Reviewer 3 - Comment #6. Thank you for your question. There is a previous systematic review and meta-analysis of whole dietary pulses which demonstrated a similar reduction in LDL-C (N=26 trial comparisons, mean difference: -0.17mmol/L, 95% CI -0.25 to -0.09). This study is discussed in lines 382-389 to demonstrate how the effects of extracted pulse proteins in the present study (N=11 trial comparisons, mean difference: -0.23mmol/L, 95% CI -0.36 to -0.10) result in a similar effect to whole pulses.
- The GRADE assessment is thorough. Could the authors elaborate on the criteria used for upgrading the evidence based on the presence of a significant dose-response gradient?
Response to Reviewer 3 - Comment #7. Thank you for highlighting the need for additional information. We have revised the description of how an upgrade would have been applied based on a dose response gradient. If a dose response was demonstrated, it could be deemed a reason for an upgrade if it supported the findings from the pooled analysis. We did not observe any dose response gradients, thus did not apply any upgrades to our GRADE assessments on any outcome. Please see lines 238-239 and in the footnote of Figure 2 for our revised description:
“The reason for upgrading the evidence was the presence of a significant dose-response gradient that aligns with the direction of the pooled effect estimate.”
- The discussion highlights the potential for extracted pulse proteins to be recommended in dietary guidelines. What are the practical implications for healthcare providers in terms of dietary advice?
Response to Reviewer 3 - Comment #8. Thank you for this important question. Our results support healthcare providers to suggest the consumption of plant-protein based foods containing extracted pulse proteins to support patient adherence to dietary recommendations. These data provide an avenue through which healthcare providers can support translation of clinical practice guidelines for obesity, diabetes and cardiovascular disease that recommend diets which emphasize consuming more plant proteins. We have revised our discussion to highlight some examples of foods they could be suggesting. Please see lines 527-528:
“The present findings support the use of extracted pulse proteins in nutrient-dense food products in alignment with dietary recommendations. These findings also support healthcare providers to translate guidelines by providing the evidence to endorse dietary advice to consume plant-based protein foods containing extracted pulse proteins for cholesterol reduction, such as plant-based burgers, ground round, or plant protein-enriched beverages.”
- The conclusion calls for more high-quality randomized trials. Are there specific types of pulse proteins or populations that should be prioritized in future research?
Response to Reviewer 3 - Comment #9. Thank you for bringing this point to our attention. We have revised the statement to provide greater detail on where there is a need for future research, which should include on chickpeas and lentils since there were no studies identified on extracted pulse proteins from these pulse types. Please see revised lines 561-563:
“To address these uncertainties, there remains a need for more large, high-quality randomized trials assessing a broader variety of pulse types, including chickpeas and lentils as there were no studies identified on extracted pulse proteins from these pulse types, as well as further exploration of effects by sex and gender. “
- Some of the figures are not labeled properly or are too small to read. I recommend revising these figures to make them more clear and readable.
Response to Reviewer 3 - Comment #10. Thank you for your feedback. We have reviewed the labelling of figures and revised accordingly. We have also revised the quality of the figures to improve clarity.
Round 2
Reviewer 3 Report
Comments and Suggestions for Authors
The study investigates the effects of extracted pulse proteins on lipid targets for cardiovascular risk reduction, as dietary pulses are recommended for managing cardiovascular disease and diabetes. A systematic review and meta-analysis of randomized controlled trials were conducted, focusing on trials with a duration of at least 3 weeks. The primary outcome was low-density lipoprotein-cholesterol (LDL-C). The analysis included seven trials with a median duration of 4 weeks and a dose of 35g/day. Extracted pulse proteins significantly decreased LDL-C by -0.23mmol/L and had similar effects on other lipid targets. Extracted pulse proteins likely result in moderate reductions in LDL-C and other lipid targets, with future studies needed to explore different types and effects by sex. The revision of the manuscript is much improved, no additional comments.